# Learning Through Limited Self-Supervision: Improving Time-Series Classification Without Additional Data via Auxiliary Tasks

## Abstract

Self-supervision, in which a target task is improved without external supervision, has primarily been explored in settings that assume the availability of additional data. However, in many cases, particularly in healthcare, one may not have access to additional data (labeled or otherwise). In such settings, we hypothesize that self-supervision based solely on the structure of the data at-hand can help. We explore a novel self-supervision framework for time-series data, in which multiple auxiliary tasks (*e.g.*, forecasting) are included to improve overall performance on a sequence-level target task without additional training data. We call this approach *limited self-supervision*, as we limit ourselves to only the data at-hand. We demonstrate the utility of limited self-supervision on three sequence-level classification tasks, two pertaining to real clinical data and one using synthetic data. Within this framework, we introduce novel forms of self-supervision and demonstrate their utility in improving performance on the target task. Our results indicate that limited self-supervision leads to a consistent improvement over a supervised baseline, across a range of domains. In particular, for the task of identifying atrial fibrillation from small amounts of electrocardiogram data, we observe a nearly 13% improvement in the area under the receiver operating characteristics curve (AUC-ROC) relative to the baseline (AUC-ROC=0.55 vs. AUC-ROC=0.62). Limited self-supervision applied to sequential data can aid in learning intermediate representations, making it particularly applicable in settings where data collection is difficult.

## 1 Introduction

Many problems involving sequential data, such as machine translation, sentiment analysis, and mortality prediction, are naturally framed as sequence-level tasks (Harutyunyan et al., 2017; Hassan et al., 2018; Radford et al., 2017). Sequence-level tasks map a sequence of observations $\mathbf{x}_{0:T}$ to a single label $y$. Learning this mapping is often made challenging due to a high-$D$ (dimension) low-$N$ (number of samples) setting (Nasrabadi, 2007). Such problems are particularly prevalent in healthcare tasks, which often involve limited quantities of labeled data captured at a high temporal resolution (*e.g.*, electrocardiogram waveforms).

In high-$D$ low-$N$ settings, researchers have had success with transfer learning techniques, by leveraging additional data to learn intermediate representations that are then used in the target task. When additional data are unavailable, it may be possible to improve the intermediate learned representation of the data with respect to the target task by considering additional tasks intrinsic to the data. In particular, we hypothesize that the structure of sequential data provides a rich source of innate supervision. For example, signal reconstruction or forecasting could improve the intermediate representation by capturing the underlying data-generating process. Such approaches are examples of self-supervision, where labels are derived from the input (as opposed to external sources).

In this paper, we show that leveraging the sequential structure of the data at-hand can lead to improved performance on sequence-level tasks (*i.e.*, the target task). More specifically, by considering self-supervised auxiliary tasks (*e.g.*, signal reconstruction), in addition to the sequence-level task, one can learn useful intermediate representations of the data. Past work investigating self-

supervision for sequential data has focused on full-signal reconstruction (Dai & Le, 2015), and to a lesser extent forecasting (Ramachandran et al., 2016). Building on past work, we examine the utility of self-supervision on sequential data when additional data are unavailable, and we propose new types of self-supervision tasks. We refer to this approach as 'limited self-supervision.' We limit the self-supervision to the data at-hand, and focus on self-supervised auxiliary tasks relevant to sequential data ordered by time (*i.e.*, time-series data).

Our main contributions are as follows:

- We demonstrate the efficacy of the proposed limited self-supervision framework for improving performance across datasets/tasks with no additional data.
- We compare the utility of several different existing forms of self-supervision in our limited-data setting, identify consistent trends across supervision types, and demonstrate the utility of combining multiple different forms of self-supervision.
- We propose a new form of self-supervision, piecewise-linear autoencoding, that trades off fine-grained signal modeling and long-term dependency propagation. We demonstrate that this is the best form of limited self-supervision across all tasks.

Our work suggests that there is a wide range of time-series and sequence classification tasks where limited self-supervision could improve performance. It also shows the value of including multiple, simultaneous streams of auxiliary self-supervision. Our findings present a methodological contribution, in the form of a useful new type of self-supervision, piecewise-linear autoencoding. Further, our empirical findings on when and how auxiliary tasks help can inform future work in developing self-supervision techniques.

## 2 RELATED WORK

Previous work has found value in self-supervised pretraining with large amounts of unlabeled data (Devlin et al., 2018; Ramachandran et al., 2016). In our work, we focus on time-series data instead of language data. Several previous works have examined self-supervision for pretraining or feature extraction in the context of time series (Hyvarinen & Morioka, 2016; Oord et al., 2018). We substitute the pretraining framework with a multitask learning framework, removing the requirement to train multiple individual models. However, in contrast to standard pretraining or multitask learning setups, we do not assume the availability of additional data for training. We posit that *even in the absence of additional data, self-supervision can lead to improved performance on the target task.*

Multitask learning deals with training a single model to perform well on multiple tasks. By simultaneously training on multiple related tasks and sharing representations, multitask models can improve generalization (Caruana, 1998). Multitask learning has been used successfully across a number of different clinical tasks (Wiens et al., 2016; Ahmed et al., 2016; Razavian et al., 2016; Harutyunyan et al., 2017). In particular, the success of multitask learning in deep learning demonstrates its value for representation learning (Ruder, 2017). Within the context of supervised learning, Schwab *et al.* considered a multitask framework for learning from sequential health data (Schwab et al., 2018). Though they used self-supervision, they only considered a setting where large amounts of unlabeled data were available.

Our work was inspired by the findings of Dai and Le (Dai & Le, 2015). Dai and Le compared sequence-autoencoding and language modeling as auxiliary tasks for leveraging large pools of unlabeled data for natural language tasks (*e.g.*, sentiment analysis). Their approach led to state-of-the-art performance on a range of problems. They found sequence-autoencoding led to larger improvements than language modeling. Interestingly, they found jointly training on the main and auxiliary task *decreased* performance relative to the baseline. In contrast, we focus on self-supervision for time series and do not assume access to additional data.

Choi et al. (2018) examines self-supervision without additional data applied to hierarchical EHR data. They also demonstrate the benefit of adding auxiliary supervision. Our work differs from this by i) using sequential as opposed to hierarchical structure, ii) examining multiple streams of simultaneous supervision and iii) comparing a broad range of auxiliary tasks on general time series data. Cao et al. (2018) find that training a sequence classification model to simultaneously forecast the data as an imputation method improves overall performance, though they suggest this is due to

better handling missing data. We examine the impact of auxiliary self-supervision more generally as a way to improve supervised representation learning.

## 3 Learning with Self-Supervised Auxiliary Tasks

In this section, we present our proposed limited self-supervision framework. After describing our notation, we present our baseline encoder-decoder architecture and describe four self-supervised auxiliary tasks.

### 3.1 Problem Definition and Notation

We define a sequence as a set of observations $\{\mathbf{x}_t\}_{t=0}^{T} : \mathbf{x}_t \in \mathcal{R}^d$ ordered by the index $t$. We denote such a sequence as $\mathbf{x}_{0:T}$. Each observation $\mathbf{x}_t$ is a $d$-dimensional vector. We focus on univariate, evenly sampled time series.

We categorize time-series tasks across three dimensions: i) target *vs.* auxiliary tasks, ii) external supervision *vs.* self-supervision, and iii) sequence-level *vs.* subsequence-level tasks. A target task is the task of interest, whereas auxiliary tasks are only useful insofar as they improve performance on the target task. External supervision occurs when the task labels are provided by an external source (*e.g.*, object recognition). Self-supervision occurs when no additional supervision is required to generate the ground truth label (*e.g.*, in autoencoding, the input itself serves as the supervision). A sequence-level task is one where the supervision pertains to the entire sequence (*i.e.*, sequence classification). A subsequence-level task provides multiple instances of supervision across the signal. In our work, all of the target tasks are sequence-level tasks requiring external supervision. We denote the label of the target task as $y$. Our auxiliary tasks are all self-supervised and may be either sequence- or subsequence-level.

### 3.2 Baseline Architecture

In this work, we compare four different self-supervised auxiliary tasks. Throughout, we consider a fixed encoder architecture, focusing on the improvements offered by the auxiliary tasks. Specifically, we focus on recurrent neural networks with LSTM cells. We use a 1-layer LSTM with a number of hidden units determined on a task-by-task basis. These architectures have proven useful for sequential tasks in many domains (Oord et al., 2016; Ramachandran et al., 2016), including health data (Harutyunyan et al., 2017). We note, however, that nothing in our framework presupposes a particular architecture, these techniques could work with any representation-learning gradient-based approach.

**Figure 1a** depicts our baseline architecture and its relation to auxiliary tasks. The encoder is implemented as a single-layer LSTM. The target decoder is a single fully connected layer mapping from hidden state $\mathbf{z}_T$ to the output. This simple output layer purposely places a heavy burden on the encoder, since the representation learned by the encoder is shared by (and thus may improve from) all tasks. We train the model by minimizing the multi-class cross-entropy between our predictions $\hat{\mathbf{y}}$ (put through a softmax activation) and the one-hot distribution representing the correct label class (**Eqn. 1**).

$$\min_{\theta_D, \theta_E} L_{target} = -\sum_i y^{(i)} \log(D(\mathbf{z}_T; \theta_D)^{(i)}) \tag{1}$$

$$\mathbf{z}_T = E(\mathbf{x}_{0:T}; \theta_E)$$

Where $\theta_E$ and $\theta_D$ are parameters for the encoder and decoder respectively and $i$ indexes class labels. We compare this baseline architecture trained with no auxiliary tasks to models trained with up to four auxiliary tasks.

### 3.3 Self-Supervised Auxiliary Tasks

We consider four self-supervised auxiliary tasks: i) autoencoding (*i.e.* reconstruction), ii) forecasting, iii) partial-signal autoencoding, and iv) piecewise-linear autoencoding (shown in **Figure 1b**). The auxiliary tasks take as input the output of the encoding network. Task $X$ is implemented as a

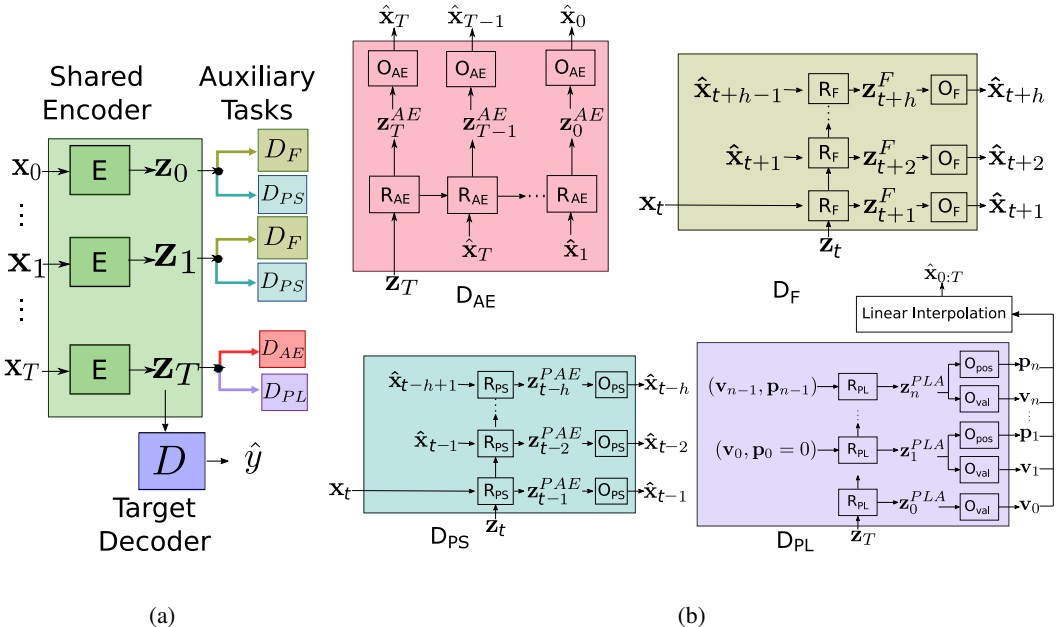

Figure 1: a) Our full task architecture. The green encoder layer is shared across all tasks. The target decoder is a fully connected layer mapping to the correct label size. The auxiliary tasks connect to hidden states. Note the sequence-level tasks only connect to the final hidden state. b) Each auxiliary task decoder $D_X$ is composed of a recurrent layer $R_X$ and an output layer $O_X$. $D_{AE}$) Autoencoder. $D_F$) Forecaster. $D_{PS}$) Partial-Signal Autoencoder. $D_{PL}$) Piecewise-Linear Autoencoder.

decoder $D_X$, composed of a recurrent layer $R_X$ and a fully connected output layer $O_X$. While our sequence-level target task requires only one prediction at the end of the sequence, some of our auxiliary tasks are subsequence-level, in which case the task takes as input the intermediate encoding representation $z_t$.

**i) Autoencoding (AE).** This is a standard method for unsupervised training in which we seek to, given the final hidden state produced by the encoder, output a reconstruction of the complete signal that minimizes error (**Eqn. 2**). This task requires that the hidden state encode compressed version of the input, compression encourages learning latent structure and discourages learning noise. The decoder, parameterized by a single-layer autoregressive LSTM, outputs a sequence: $D_{AE}(z_T) = \hat{x}_{0:T}$. Note $z_T = E(x_{0:T}; \theta_E)$, and thus depends on $\theta_E$.

$$\min_{\theta_E, \theta_{AE}} L_{AE} = ||x_{0:T} - D_{AE}(z_T; \theta_{AE})||_2^2 \tag{2}$$

**ii) Forecasting.** In this task, the decoder takes the hidden state produced at any time step by the decoder and attempts to predict the next $h$ (for example, 6) elements in the sequence. This requires the hidden state to encode the dynamics of the data-generating process. The decoder outputs $x_{t+1:t+h}$ given past values $x_{0:t}$, minimizing (**Eqn. 3**). We use a single-layer LSTM similar to the AE decoder, though it is used at each time-step to decode the next $h$ observation values.

$$\min_{\theta_E, \theta_F} L_F = \sum_{t=1}^{T-h} ||x_{t+1:t+h} - D_F(z_t; \theta_F)||_2^2 \tag{3}$$

$$z_t = E(x_{0:t}; \theta_E)$$

**iii) Partial-Signal Autoencoding (PS-AE).** This auxiliary task is a variant of AE that differs in three ways: 1) instead of decoding the full signal $x_{0:T}$ it decodes only the previous $h$ (for example, 6) steps of the signal $x_{t-h:t-1}$, 2) instead of one prediction being made at the end of the encoder, a prediction is made at every encoding step from $x_h$ onwards, and 3) the input to the decoding layer includes the current value, $x_t$ (**Eqn. 4**). It is implemented identically to the Forecast Decoder, the only difference is that it predicts the previous $h$ observation values. This task allows us to examine

the impact of signal reconstruction without requiring learning long-term dependencies, allowing for a more meaningful comparison with Forecasting.

$$\min_{\theta_E, \theta_{PS}} L_{PS} = \sum_{t=1}^{T-h} ||\mathbf{x}_{t-h:t-1} - D_{PS}(\mathbf{z}_t; \theta_{PS})||_2^2 \tag{4}$$

**iv) Piecewise-Linear Autoencoding (PL-AE).** A piecewise-linear approximation, or a combination of line segments, is capable of efficiently representing a wide class of signals (particularly non-periodic signals). It is a promising choice for an auxiliary task as it encourages a compact representation capturing the most important details of the signal. A piecewise-linear representation consists of two length $n+1$ vectors (where $n$ is the number of linear segments in the signal), a value vector $\mathbf{v}$ and a position vector $\mathbf{p}$. The positions are defined as proportions of the original signal length, between 0 and 1. The decoder produces a series of points which define a piecewise-linear reconstruction of the complete input signal. The reconstruction is defined by linear interpolation between the series of points $(p_0, v_0) \dots (p_n, v_n)$, where $v_i$ is the value of the signal at the point $i$, and $p_i$ is the relative position (or time) where the point occurs. Our decoder produces $n+1$ such points using a single-layer autoregressive LSTM where the hidden state is fed to two output layers which map $\mathbf{z}_i^{PL} \to (v_i, p_i)$. For each point $i \in [0, n+1]$ we feed the previous point value $v_{i-1}$ and the sum of previous positions $\sum_{j=0}^{j=i-1} p_j$ to the decoder. After we have generated the target number of points, we normalize the position values to enforce $\sum_{j=0}^{n+1} p_j = T$ and perform the interpolation. As this entire process is differentiable, we optimize the decoder directly on the interpolated reconstruction loss (**Eqn. 5**).

$$\min_{\theta_E, \theta_{PL}} L_{PL} = ||\mathbf{x}_{0:T} - D_{PL}(\mathbf{z}_T; \theta_{PL})||_2^2 \tag{5}$$

Additional details on these tasks can be found in **Appendix A.1**.

## 3.4 TRAINING

We optimize our model to minimize the loss:

$$L = L_{target} + L_{Aux} \tag{6}$$

Where $L_{target}$ is a cross-entropy loss and

$$L_{Aux} = \alpha_{AE} L_{AE} + \alpha_F L_F + \alpha_{PS} L_{PS} + \alpha_{PL} L_{PL} \tag{7}$$

is a weighted summation of auxiliary MSE losses. The weighting terms $\alpha_X$ are defined as 0 if the auxiliary task $X$ is not being used for training. If the task is being used, then $\alpha_X = \frac{L_{target}}{L_X}$, where the losses are calculated at the beginning of training using the newly-initialized network on the training data. This ensures that all tasks have losses of similar magnitude.

## 4 EXPERIMENTAL SETUP

To test our hypothesis, that auxiliary self-supervision improves performance on sequence-level tasks, we consider at a variety of sequence-level tasks across different types of synthetic and real data.

## 4.1 TARGET TASKS & DATASETS

We consider the following three tasks (two of which are based on publicly available real datasets):

**Piecewise-Linear Segment Prediction (PLA).** We begin with simulated data, as it allows us to estimate the ability of self-supervised auxiliary tasks to identify long-term dependencies in the data. The dataset is composed of piecewise-linear signals, each of length 100. Point values are drawn from a uniform distribution and lie between -1 and 1. The number of line segments also varies uniformly between 1 and 6. The target task for this dataset is to estimate the number of distinct segments that occurred in the signal. 1,000 training, validation, and test sequences were generated independently.

**Patient Classification using Glucose Data (T1D).** This task uses publicly available continuous glucose monitor (CGM) data collected from people with type 1 diabetes (T1D) (Fox et al., 2018).

Each signal $\mathbf{x}_{0:T}$ consists of 288 glucose measurements sampled every five minutes over the course of a day. This dataset contains 1,863 days of data from 40 patients. In this task, we aim to classify patients based on their data. Here, $y \in \{1, \ldots, 40\}$ represents the patient. Classifying patients is a proxy for the important problem of identifying signal dynamics. We preprocess the data by removing physiologically implausible glucose measurements, and linearly interpolating missing chunks of data. We exclude signals in which more than 20% of the measurements are missing and those that are missing a contiguous block longer than two hours. Data were collected by a series of multi-day sessions separated by three-month intervals. As our test set, we consider the final recording session from each patient. We select our validation set randomly from the remaining data.

**Atrial Fibrillation Detection (AF).** Our final task uses electrocardiogram (ECG) data from the publicly available 2017 PhysioNet Challenge (Clifford et al., 2017), in which the goal was to automatically diagnose atrial fibrillation (AF). This dataset contains four unevenly distributed classes: normal sinus rhythm, AF, other arrhythmia, and noise. We use the training data provided for the competition (the test data are not publicly available), resulting in 8,528 samples. 771 of those signals are labeled AF. We exclude signals with less than 30 seconds of data (967 signals total, 127 with AF) and truncate all signals to exactly 30 seconds. We also downsample the data, reducing signal size from 9,000 to 125. This speeds up training time and eases memory requirements. We use the validation set provided for the challenge as the test set (removing those examples from the training set), and randomly sample 10% of the training data for use as a validation.

### 4.2 IMPLEMENTATION DETAILS

We implement all models in PyTorch (Paszke et al., 2017), and optimize model parameters using Adam (Kingma & Ba, 2014) with an initial learning rate of $1e - 3$ (the default PyTorch value). In practice we found altering the decoding horizon had little effect on performance, so we used $h = 6$ for all experiments. All encoding/decoding layers are composed of a single recurrent layer an identical number of hidden units, set on a per-task basis using performance on the validation set to balance training time, memory constraints, and target-task performance (evaluated using the early-stopping validation set). We used 128 hidden units for the PLA target task, 512 hidden units for the T1D target task, and 256 hidden units for the AF target task. The decoding layers also have 1 (or two for PL-AE) fully connected output layers. For the PL-AE, we set the number of line segments $n = 6$ as a reasonable size to approximate many signal types. We mitigate the risk of overfitting by using early stopping on a withheld validation set, training until we fail to improve performance for over 50 epochs and reporting test performance for the best performing model on the validation data. Some auxiliary tasks make predictions at multiple points in the signal. This helps prevent the vanishing gradient problem, which can impede learning with large sequences. To avoid conflating these sorts of improvements with those caused by learning better representations, we use label propagation with our target task (sequence classification), linearly annealing contributions to the loss function over the length of the signal (Dai & Le, 2015). The propagated losses are combined using a weighted average, with the weights linearly annealed from 0 to 1 over the length of the signal. Our code and synthetic data will be made publicly available to allow for replication and extension. For the purposes of double-blind peer review, we have released the code and PLA data on an anonymous Google Drive account[1].

## 5 RESULTS

We begin by establishing that the proposed self-supervised auxiliary task framework improves target task performance. We then look more carefully at the effects of different types of auxiliary tasks. We conclude by looking at the relationship between auxiliary task and target task performance, which sheds light on the mechanism by which auxiliary tasks improve performance.

To evaluate the results of these experiments, we measure the macro-averaged AUC-ROC on the target task, and the mean absolute percent error (MAPE) for the auxiliary tasks. We repeat all experiments using three random initializations and average the results.

**The Benefit of Limited Self-Supervision.** We begin by examining our main hypothesis, that limited self-supervision improves sequence-level task performance without additional data. In **Figure 2**,

---

[1]https://bit.ly/2l2X4ax

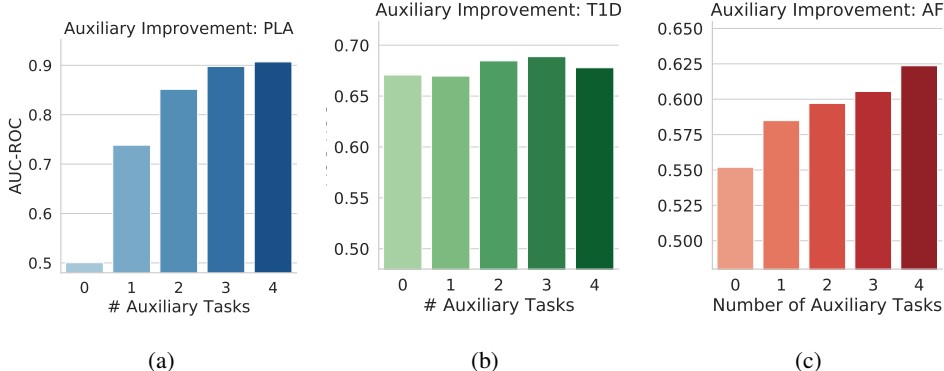

(a)          (b)          (c)

Figure 2: Performance across three target tasks by number of auxiliary tasks used (averaged over all possible orderings). In general, we observe that the greater the number of auxiliary tasks, the greater the performance for all three tasks. The marginal improvement from including additional auxiliary tasks appears to taper off as the number of tasks increases.

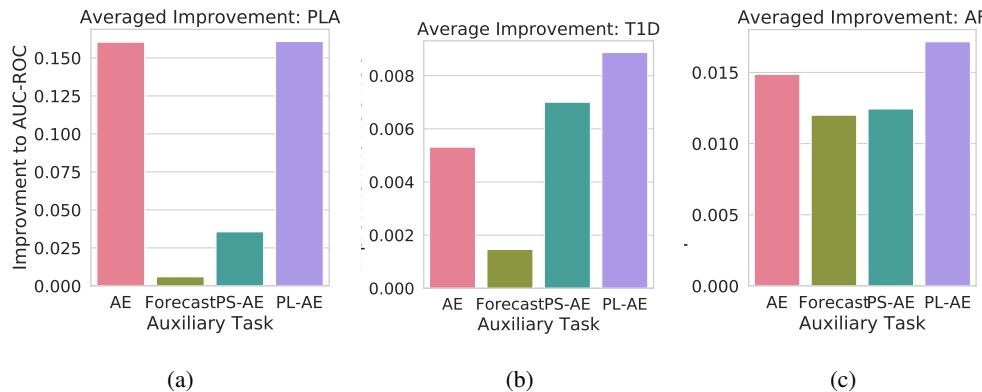

(a)          (b)          (c)

Figure 3: The average marginal contribution of each auxiliary task across target tasks. The PL-AE task consistently offers the greatest marginal improvement, and the forecasting task consistently provides the least.

we plot target-task performance across our three tasks with a varying number of self-supervised auxiliary tasks. We estimate performance for a given number of auxiliary tasks by averaging the performance of all possible combinations. For all three sequence-level tasks, the inclusion of four auxiliary tasks improves performance relative to no auxiliary tasks. Moreover, we observe that performance tends to increase with the number of auxiliary tasks. The one exception is in the T1D task, where improvement peaks at three auxiliary tasks.

**Relative Contribution of Different Auxiliary Tasks.** We now examine how different auxiliary tasks contribute differently to performance. To measure the impact of individual auxiliary tasks, we average performance across all models that include the auxiliary task versus all models that do not. This allows us to observe an auxiliary task's individual contribution, and its ability to usefully combine with other streams of self-supervision. The averaged change in AUC-ROC indicates the marginal improvement offered by the task (**Figure 3**). PL-AE outperforms all other auxiliary tasks, including AE, on all three datasets. Since PL-AE limits the temporal granularity of the output, this suggests that modeling fine temporal granularity does not help, and may even hurt performance. The forecasting task underperforms all other auxiliary tasks. This finding is in line with the findings of (Dai & Le, 2015). However, the explanation Dai and Le provide (that the AE encourages long-term dependency modeling) is inconsistent with the performance of PS-AE, which also does not model long-term dependencies. The fact that the PS-AE outperforms the forecasting task, but generally underperforms AE, suggests that while modeling long-term dependencies may improve performance,

Table 1: The performance of particular combinations of auxiliary tasks. All methods include the dataset-specific target task. AE and Forecast refers to Autoencoding and Forecasting respectively, the auxiliary tasks explored in (Dai & Le, 2015). PLAE and PSAE refer to Piecewise-Linear Autoencoder and Partial-Signal Autoencoder, our novel forms of self-supervision. We see that our newly proposed forms of self-supervision outperform the other approaches on all datasets.

| | AUC-ROC * 100 | | |
| --- | --- | --- | --- |
| Auxiliary Tasks | PLA | T1D | AF |
| None | $50.0 \pm 0.4$ | $67.1 \pm 1.3$ | $55.2 \pm 4.0$ |
| AE | $88.6 \pm 0.3$ | $66.7 \pm 0.4$ | $59.8 \pm 1.2$ |
| Forecast | $53.5 \pm 5.2$ | $66.3 \pm 0.5$ | $57.0 \pm 2.0$ |
| AE+Forecast (Dai & Le, 2015) | $89.5 \pm 0.6$ | $67.4 \pm 1.0$ | $61.0 \pm 0.7$ |
| PLAE+PSAE **(Ours)** | $89.6 \pm 0.4$ | $\mathbf{69.6 \pm 0.5}$ | $60.0 \pm 3.3$ |
| All **(Ours)** | $\mathbf{90.7 \pm 0.3}$ | $67.8 \pm 0.7$ | $\mathbf{62.4 \pm 2.1}$ |

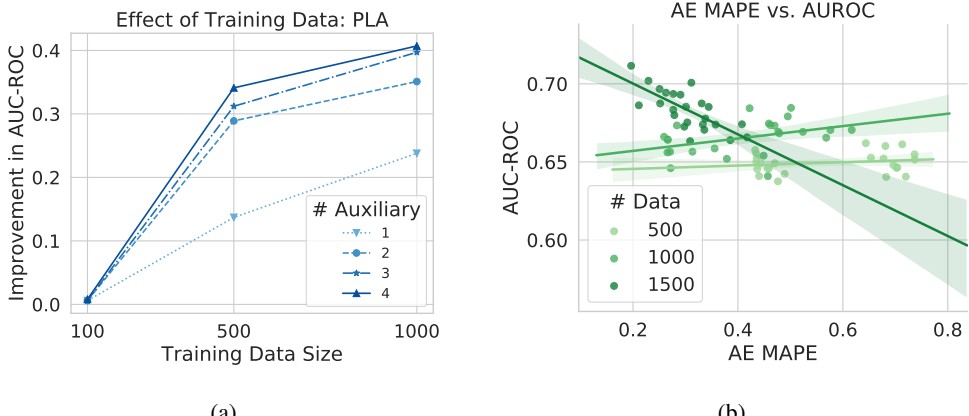

(a)  (b)

Figure 4: A) The effect training data size has on auxiliary tasks. As the amount of data increases, we tend to see an increase in the improvement afforded by the auxiliary tasks. B) The relationship between auxiliary task performance (in MAPE, lower is better) and target task performance (in AUC-ROC, higher is better) on the T1D data. Calculated over all training data sizes, there is a somewhat weak relationship (Pearson R = -0.53, line not shown). However, when we condition on the amount of training data, we find that different relationships emerge (Pearson R = 0.34 when training size = 1,000 vs. -0.72 when training size = 1,500).

there is likely some other reason that AE works well. We also examine the performance of particular combinations of self-supervised auxiliary tasks in **Table 2**. We see that our proposed auxiliary tasks outperform and are complementary to standard auxiliary task combinations.

***Why* do Self-Supervised Auxiliary Tasks Help?** To investigate the underlying mechanism by which auxiliary tasks improve performance, we examine model performance as we vary the number of auxiliary tasks and the amount of training data (results shown in **Figure 4a**). As the amount of training data increases, the added value from the auxiliary tasks increases on average. This suggests that auxiliary tasks are not simply acting as a form of regularization, since otherwise we would expect to see larger improvements on smaller training sets.

If the auxiliary tasks result in better representations (not just regularized representations), their impact on performance should correlate with the auxiliary task performance. A *decrease* in auxiliary task error should lead to a better intermediate representation and an *increase* in target task performance (*i.e.*, AE MAPE and AUC-ROC should be negatively correlated). Meanwhile, if the auxiliary tasks work as regularizers, exhausting representation capacity, we would expect to see little effect or a positive correlation (if the regularization effect is too strong, and auxiliary performance comes at the *cost* of target performance).

We explore this, specifically for AE with the T1D data in **Figure 4b**. This relationship is highly dependent on the amount of data. When training data are limited, there is a weak positive correlation between auxiliary task error and target performance, suggesting a regularizing effect. However, with

the full amount of training data, we see a strong negative correlation. Auxiliary tasks give sizable improvements at 1,000 and 1,500 training examples (averaged improvement of 0.016 and 0.010 respectively). This suggests that 1) auxiliary tasks act both to regularize *and* to enhance intermediate representations, depending on the amount of data, and 2) there is an amount of training data where they are effective in either role. These findings suggest that auxiliary self-supervised tasks may be useful across a wide range of training set sizes.

## 6 CONCLUSIONS

In this paper, we introduced a limited self-supervised framework, in which we sought to improve sequence-level task performance without additional data. By jointly training our target task with auxiliary self-supervised tasks, we demonstrated small but consistent improvements across three different sequence classification tasks. Our novel piecewise-linear autoencoding task emerged as the most useful auxiliary task across all datasets. In contrast, forecasting, which presents an intuitively appealing form of self-supervision, led to the smallest improvements.

Across a range of training set sizes, we showed that the value of auxiliary tasks lies in improving intermediate representations learned by the network. When limited training data are available, these tasks serve as a form of regularization. With more training data, performance on the auxiliary tasks improves and so does performance on the target task, suggesting more useful intermediate representations. In the context of time-series analysis, limited self-supervision is an effective form of supervision and comes at little cost. Going forward, researchers seeking to improve performance on sequence-level target tasks should consider incorporating self-supervised auxiliary tasks.

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

# A APPENDIX

## A.1 AUXILIARY TASKS: ADDITIONAL MOTIVATION AND DETAILS

### A.1.1 AUTOENCODING

*Motivation.* Autoencoding encourages the hidden state to retain all relevant information from the original input. By reducing the size of the hidden state, our model must learn a compressed representation of the signal. Compression encourages learning latent structure and discourages learning noise.

*Details.* We use a single-layer autoregressive LSTM for the recurrent layer $R_{AE}$, which is initialized using the hidden and cell states from the encoder LSTM and sequentially decodes hidden states $\mathbf{z}_t^{AE}$. These hidden states are fed into a fully-connected output layer, $O_{AE}$, that maps $\mathbf{z}_t^{AE} \rightarrow \hat{\mathbf{x}}_t$. The output $\hat{\mathbf{x}}_t$ is then fed into $R_{AE}$, generating $\mathbf{z}_{t+1}^{AE}$, continuing until $\hat{\mathbf{x}}_{0:T}$ is fully generated. To provide a shorter path for gradient flow, we decode in reverse order, generating $\hat{\mathbf{x}}_{T:0}$ instead of $\hat{\mathbf{x}}_{0:T}$ (Goodfellow et al., 2016).

### A.1.2 FORECASTING

*Motivation.* Forecasting encourages $E$ to encode the dynamics of the data-generating process. Without information about the underlying dynamics, future value prediction is either challenging or trivial (if the signal does not change). Such dynamics may carry valuable information for a range of sequence-level tasks, though the aspects of the dynamics most relevant to the target task may differ from those that predict future values.

*Details.* We focus on a multi-output forecasting architecture, predicting several future values simultaneously (Taieb et al., 2010). This is done using a recurrent decoder, similar to the autoencoder, but applied at each encoding step and expanded only $h$ steps, where $h$ is a hyperparameter.

### A.1.3 PARTIAL-SIGNAL AUTOENCODING

*Motivation.* Previous work has found advantages to reconstruction over prediction (Dai & Le, 2015). Dai and Le hypothesize the observed superiority of sequence autoencoding over forecasting may result from the short-term nature of the language modeling task (only predicting the next word). To investigate the effect of short-term dependency auxiliary tasks, we use a multi-step forecasting system, where we can examine the effect of varying the prediction horizon. Analogously, we use PS-AE to examine the effect of producing short-term reconstructions.

*Details.* PS-AE is implemented using a setup identical to our forecasting approach, except we estimate the *previous* $h$ values instead of the *subsequent* values (**Figure 1 a**, $D_{PS}$).

### A.1.4 PIECEWISE-LINEAR AUTOENCODING

*Motivation.* AE encourages fine-grained modeling of the signal over long periods, whereas PS-AE encourages fine-grained modeling over a short period. While PS-AE can control the range of temporal dependencies modeled using the decoding horizon $h$, it does not vary the granularity at which the output is modeled. What level of signal granularity is required for reasonable target-task performance? To explore this question, we introduce Piecewise-Linear (PL) Autoencoding.

*Details.* To generate a PL representation of a signal with $n$ distinct pieces, we take the encoded representation of the signal, $\mathbf{z}_T$, and feed it into a recurrent layer $R_{PL}$. **Figure 1 a**, $D_{PL}$ illustrates our point-generation system. The sequential output of the decoder, $\mathbf{z}_j^{PL}$, generates the $j^{\text{th}}$ point value and position. Specifically, we use two fully-connected output layers, $O_{val}$ and $O_{pos}$, that map $\mathbf{z}_j^{PL} \rightarrow v_j, p_j$ respectively. Since we know $p_0 = 0$, at the first step we generate only an initial value. We continue the decoding process, feeding $\mathbf{v}_j, \sum_{k=0}^{j} \mathbf{p}_k$ to the decoder to generate $\mathbf{z}_{j+1}^{PL}$, until we have generated $n + 1$ points. We then use linear interpolation to map $(\mathbf{v}_0, \mathbf{p}_0) \dots (\mathbf{v}_n, \mathbf{p}_n) \rightarrow \hat{\mathbf{x}}_{0:T}$. We normalize the position vector and use the cumulative summation to determine segment positions.

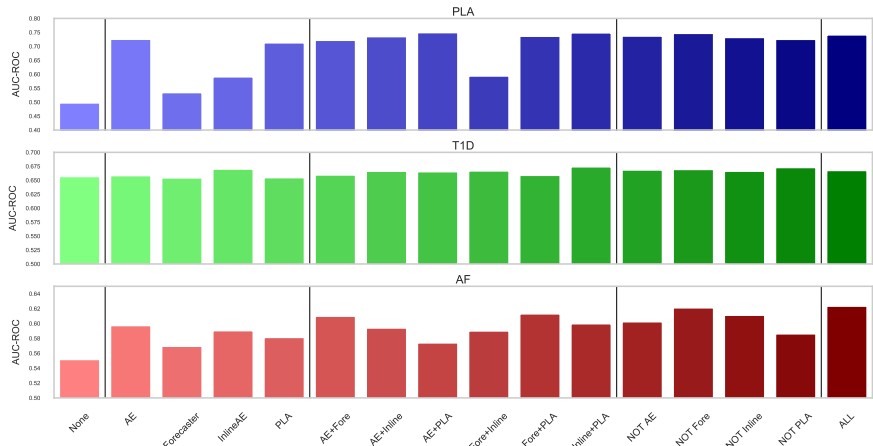

Figure 5: Average AUC-ROC for every combination of Auxiliary task for all three analyses. Here, columns marked 'NOT' included all but the indicated auxiliary task.

Table 2: Accuracy on 7 data sets from the UCR Archive.

| Data Set | Baseline-ACC | Best-ACC | Best-Tasks | SOTA- ACC |
|---|---|---|---|---|
| ElectricDevices | 57.5 | 59.6 | Forecaster | 89.5 |
| FordB | 60.3 | 75.4 | Inline + PLA | 92.9 |
| FordA | 61.4 | 86.6 | Not Forecaster | 96.5 |
| Wafer | 89.1 | 95.2 | AE | 100.0 |
| HandOutlines | 54.7 | 54.7 | None | 92.4 |
| TwoPatterns | 64.4 | 100 | Forecaster+Inline | 100 |
| StarLightCurves | 84.6 | 89.0 | Not PLA | 98.0 |

## A.2 ALL COMBINATIONS OF AUXILIARY TASKS

In Figure 5 we present plots showing average AUC-ROC for all possible combinations of auxiliary tasks, for each data set. Here the contribution of each task can be examined.

## A.3 APPLICATION TO UCR TIMES SERIES DATA-SETS

In order to evaluate the use of self-supervised auxiliary tasks in an easily replicable setting, and in order to facilitate comparisons with state-of-the are methods, we performed classification on 7 data sets from the UCR Time Series Archive https://www.cs.ucr.edu/~eamonn/time_series_data_2018/. We included tasks that had fixed length sequences and a sample to label ratio greater than 100. We only included the subset of these tasks for which our baseline architecture performed better than random chance. These selections were to ensure that the tasks were reasonably well suited to our baseline architecture. For each dataset, the last 20% of the training sample was used as a validation set, with the testing sample used as a test set.

Table 2 shows baseline results for each task (accuracy when our model was run with no auxiliary tasks), the highest accuracy achieved with any combination of auxiliary tasks, and state-of-the-art performance according to timeseriesclassification.com. Although the baseline architecture is routinely and drastically outperformed by state-of-the-art methods, the addition of auxiliary tasks improved performance to be significantly closer to the state-of-the-art level in most cases, and our model achieved at- state-of-the-art accuracy for the Two Patterns task. Although our model was unable to achieve state-of-the-art performance on most tasks, the improvement over baseline indicates that the addition of self-supervised auxiliary tasks could enhance better suited baseline architectures on these and other tasks.

