# OpenReview forum: "Learning Through Limited Self-Supervision: Improving Time-Series Classification Without Additional Data via Auxiliary Tasks"
_ICLR.cc/2020/Conference — Reject_

### Official Review · AnonReviewer2 · 2019-10-21
**Official Blind Review #2**

**Rating:** 1

**Review:**

This paper proposes a so called self-supervised method for learning from time series data in healthcare setting. Specifically, here self-supervision is achieved via designing auxiliary tasks based on data's internal structure to create more labeled auxiliary training tasks.

From both perspectives of methods and applications, the proposed model has very limited novelty. It is just one application of multitask learning. Also very similar idea has been implemented by [1]. In [1], the authors learn multi-level embedding to make disease/risk prediction, where the embedding was jointly trained by performing auxiliary prediction tasks that rely on this inherent EHR structure. The authors need to state what is the novelty of the proposed method compared with [1].

In addition, the performance evaluation missed many baselines. Table 1 seems more like a ablation study rather than a performance comparison. You need to compare with all state-of-the-art models in computational phenotyping in order to show the performance advantage brought by the proposed mode design.

[1] Edward Choi, Cao Xiao, Walter Stewart, Jimeng Sun, MiME: Multilevel Medical Embedding of Electronic Health Records for Predictive Healthcare,  NeuRIPS, 2018



**Experience Assessment:**

I have published in this field for several years.

**Review Assessment: Checking Correctness Of Derivations And Theory:**

I carefully checked the derivations and theory.

**Review Assessment: Checking Correctness Of Experiments:**

I carefully checked the experiments.

**Review Assessment: Thoroughness In Paper Reading:**

I read the paper thoroughly.

---

> ### Author Response · Authors · 2019-11-15
> **Response to Review 2**
>
> Thank you for your thorough review.
>
> In response to your concerns:
>
> - Novelty of the proposed method compared with [1]:
> Limited self-supervision uses a multitask framework that, critically, requires no external labels to improve accuracy on a single task. Most applications of multitask learning require additional ‘external’ labels, but our method is applicable even in situations where such labels aren’t available, hence its novelty.
>
> Although [1] also proposes a limited self-supervised framework, their method is applicable only in the EHR setting, since it requires additional labels from diagnosis and treatment codes. Still, we agree this is related, and have updated Section 2 to include a discussion of it. The relative novelty of our work is to examine limited self-supervision on general time-series tasks with a variety of different auxiliary tasks. In particular, we examine the relative merits of different auxiliary tasks, propose a novel auxiliary task (PLAE), and show the importance of including multiple forms of auxiliary supervision.
>
> - Insufficient baselines:
> Our main goal was not to obtain state-of-the-art results on computational phenotyping tasks, but to investigate the utility of a limited self-supervision framework to sequence classification. To this end we compared to a fully supervised network (our baseline), and ran experiments investigating different types of auxiliary tasks. In order to present this frameworks applicability to a broader array of datasets, we have added an analysis of 7 datasets from the UCR repository (see Section A3 in the supplement). We showed that the addition of self-supervised auxiliary tasks offered sizable improvements over our baseline architecture on most datasets. Although we achieved state-of-the-art level performance on only one dataset,  we find the consistent improvement in performance over the baseline indicates the general promise of this approach.

---

### Official Review · AnonReviewer1 · 2019-10-22
**Official Blind Review #1**

**Rating:** 3

**Review:**

This paper propose an approach for self-supervised learning on time series.
Three datasets are considered (simulation and 2 healthcare datasets).
The gist of the contribution is to both optimize prediction loss
of the true task and at the same time do a good job for a family
of auxiliary tasks. 4 auxiliary tasks are considered. While
the first 3 auxiliary tasks are quite common, the 4th tasks
called piecewise-linear autoencoding appears novel. The idea
is that the hidden representation of the LSTM should be a good predictor
of the past using a piecewise-linear approximation.
The author coin the term "limited self-supervision" for their approach
although it's not clear why it is fundamentally not just self-supervised
learning as it has been proposed in the past.

The paper is overall well written and addresses the relevant issue
of learning from limited annotated data.

Major concerns

- It is yet another way to do self-supervised learning on time series
and no clear benchmark with alternatives is provided (time contrastive
learning (TCL) or Contrastive Predictive Coding (CPC) https://arxiv.org/pdf/1807.03748.pdf
etc.)

- On any of the applied problem it is not clear if the proposed
approach brings an improvement on the state-of-the-art or if it's
just an illustration of the method disconnected from the literature
of the application.


**Experience Assessment:**

I have published one or two papers in this area.

**Review Assessment: Checking Correctness Of Derivations And Theory:**

I did not assess the derivations or theory.

**Review Assessment: Checking Correctness Of Experiments:**

I assessed the sensibility of the experiments.

**Review Assessment: Thoroughness In Paper Reading:**

I read the paper thoroughly.

---

> ### Author Response · Authors · 2019-11-15
> **Response to Reviewer 1**
>
> Thank you for your review. We were glad that you found our paper well written and problem relevant.
>
> In response to your concerns:
>
> - The difference between self-supervision and limited self-supervision:
> Works that use self-supervision generally assumes the availability of large amounts of unlabeled data that can be used for pretraining or feature extraction. In contrast, we do not make this assumption - hence the term ‘limited.’ We show that even when all of your data is supervised, self-supervision can still improve performance relative to a pure supervised baseline.
>
> - No clear benchmark with alternatives is provided (such as TLC or CPC):
> The main contribution of this work is to examine self-supervised learning in a setting without additional unlabeled data, where we instead extract additional supervision from the sequential structure inherent in the labeled data. We have explored the efficacy of various common forms of self-supervision on this broadly applicable setting, and proposed a novel form of self-supervision for this setting. The inclusion of additional forms of self-supervision, such as TCL or CPC could serve as an interesting direction for future work. We have updated our paper to mention these papers.
>
> - Does the approach achieve state-of-the-art results?
> n two of our three sequence classification tasks (PLA and T1D) we are unaware of other published evaluations we could compare to. On the AF task, our results are not state-of-the-art. Our main goal was not to obtain state-of-the-art results on these sequence level classification tasks, but to investigate the utility of a limited self-supervision framework. In order to present this frameworks applicability to a broader array of datasets, we have added an analysis of 7 datasets from the UCR repository (see Section A3 in the supplement). We showed that the addition of self-supervised auxiliary tasks offered sizable improvements over our baseline architecture on most datasets. Although we achieved state-of-the-art level performance on only one dataset,  we find the consistent improvement in performance over the baseline indicates the general promise of this approach.

---

### Official Review · AnonReviewer3 · 2019-10-23
**Official Blind Review #3**

**Rating:** 1

**Review:**

This paper proposes the use of many auxiliary tasks to boost the performance on a target task by means of `'self-supervision'. Specifically, they considered auto-encoding, forecasting, partial-segment auto-encoding, and piecewise-linear auto-encoding.

There are major concerns that should be clarified or described in detail.
1) The overall architecture is not complete. the architectures used in the experiments are not described concretely.
2) To this reviewer, the idea of self-supervision is similar to the unsupervised learning for representation learning.
3) The methods of BERT (Bidirectional Encoder Representations from Transformers) [Devlin et al., 2018] or BRITS (Bidirectional Recurrent Imputation for Time Series) [Cao et al., 2018], although different for their target tasks in their original work, could be also regarded as self-supervision technique and could be interesting to compare with them.
4) The experimental settings are not described well, thus lack of reproducibility
5) It is unclear which aux-tasks were applied in Fig. 2. Further to better understand and analyze the results, it is required to conduct more rigorous ablation studies.
6) There is no comparison with recent work on the same datasets.

**Experience Assessment:**

I have read many papers in this area.

**Review Assessment: Checking Correctness Of Derivations And Theory:**

N/A

**Review Assessment: Checking Correctness Of Experiments:**

I assessed the sensibility of the experiments.

**Review Assessment: Thoroughness In Paper Reading:**

I read the paper at least twice and used my best judgement in assessing the paper.

---

> ### Author Response · Authors · 2019-11-15
> **Response to Reviewer #3**
>
> Thank you for your thorough review. In response to your major concerns:
>
> 1) We have uploaded a revised version of the paper where we provide more complete and concrete descriptions of our architecture.
>
> 2) We believe self-supervision can be viewed as an unsupervised learning approach for representation learning. We focus on a setting, limited self-supervision, that does not assume access to additional unlabeled data. One of the major contributions of our work is that we demonstrate self-supervision is useful in such situations.
>
> 3) BERT is designed to pre-train deep bidirectional representations from unlabeled text by jointly conditioning on both left and right context in all layers. It pre-trains on two tasks: next-sentence prediction and hidden token identification, but these tasks are somewhat specific to NLP. We focus on tasks that are applicable to time-series. Moreover, our setup isn’t a pre-training setup, in which one assumes a large amount of additional unlabeled data, but seeks to leverage the data at hand.
>
> BRITS aims to impute missing values while solving the sequence level task (e.g., predict in-hospital mortality). There are parallels between their work and ours, notably their RITS model is functionally equivalent to our Baseline+Forecasting model. However, our main goal was to improve the representation learned by a network on fully observed and supervised data.
>
> 4) In our revised draft of the paper, we have attempted to better describe our experimental settings. Additionally, we have posted all of our code to an anonymous google drive account, we will upload this to the authors github account after the review process has concluded. Since all datasets we report results on are publicaly available and our source code is publicly available, we believe our results are reproducible.
>
> 5) In Figure 2 we use all auxiliary tasks averaging across subsets. For example when # of auxiliary tasks =1 this corresponds to an average of the performance of all single-auxiliary task models. When # of auxiliary tasks=2 we consider all combinations of size 2 and average the resulting performance. This figure demonstrates that additional streams of self-supervision tends to help, as average performance increases with number of auxiliary tasks. To provide additional insight into the performance of all auxiliary task combinations, we have added the full results (not just the averages) in the supplement section A2.
>
> 6) Our main goal was not to obtain state-of-the-art results on these sequence level classification tasks, but to investigate the utility of a limited self-supervision framework. To this end we compared to a fully supervised network (our baseline), and ran experiments investigating different types of auxiliary tasks. In order to present this frameworks applicability to a broader array of datasets, we have added an analysis of 7 datasets from the UCR repository (see Section A3 in the supplement). We showed that the addition of self-supervised auxiliary tasks offered sizable improvements over our baseline architecture on most datasets. Although we achieved state-of-the-art level performance on only one dataset,  we find the consistent improvement in performance over the baseline indicates the general promise of this approach.

---

### Author Response · Authors · 2019-11-15
**Summary of our Main Contribution**

Though others have previously demonstrated that self-supervision can be used to learn useful discriminative representations from large pools of unlabeled data, we show that when used as auxiliary tasks, self-supervision can improve the representation learning without any additional data. To the best of our knowledge, we are the first to propose limited self-supervision as a general framework for representation learning. To this end, we investigated this framework by: 1) considering a range of different auxiliary tasks and 2) exploring the effect of combining these tasks. Based on comparisons across a range of datasets, we find that multiple simultaneous streams of auxiliary self-supervision improve performance over a single stream.

---

### Decision · Program_Chairs · 2019-12-19

**Decision:**

Reject

**Comment:**

The paper addresses  an important problem of self-supervised learning in the context of time-series classification. However, all reviewers raised major concerns regarding the novelty of the approach and the quality of empirical evaluation, including insufficient comparison with the state-of-art and reproducibility issues. The reviewers agree that the paper, in its current state, does not path the ICLR acceptance threshold, and encourage the authors to improve the paper based on the provided suggestions.